# Identification of Short-Rotation Eucalyptus Plantation at Large Scale Using Multi-Satellite Imageries and Cloud Computing Platform

**Xinping Deng, Shanxin Guo, Luyi Sun**  **and Jinsong Chen \***

Shenzhen Institutes of Advanced Technology, Chinese Academy of Sciences, Shenzhen 518055, China;
xp.deng1@siat.ac.cn (X.D.); sx.guo@siat.ac.cn (S.G.); ly.sun@siat.ac.cn (L.S.)
**\*** Correspondence: js.chen@siat.ac.cn; Tel.: +86-755-86392370

**Abstract:** A new method to identify short-rotation eucalyptus plantations by exploring both the changing pattern of vegetation indices due to tree crop rotation and spectral characteristics of eucalyptus in the red-edge region is presented. It can be adopted to produce eucalyptus maps of high spatial resolution (30 m) at large scales, with the use of open remote sensing images from Landsat 8 Operational Land Imager (OLI), MODerate resolution Imaging Spectroradiometer (MODIS), and Sentinel-2 MultiSpectral Instrument (MSI), as well as a free cloud computing platform, Google Earth Engine (GEE). The method is composed of three main steps. First, a time series of Enhanced Vegetation Index (EVI) is constructed from Landsat data for each pixel, and a statistical hypothesis testing is followed to determine whether the pixel belongs to a tree plantation or not based on the idea that tree crops should be harvested in a specific period. Then, a broadleaf/needleleaf classification is applied to distinguish eucalyptus from coniferous trees such as pine and fir using the red-edge bands of Sentinel-2 data. Refinements based on superpixel are performed at last to remove the salt-and-pepper effects resulted from per-pixel detection. The proposed method allows gaps in the time series that are very common in tropical and subtropical regions by employing time series segmentation and statistical hypothesis testing, and could capture forest disturbances such as conversion of natural forest or agricultural lands to eucalyptus plantations emerged in recent years by using a short observing time. The experiment in Guangxi province of China demonstrated that the method had an overall accuracy of 87.97%, with producer's accuracy of 63.85% and user's accuracy of 66.89% for eucalyptus plantations.

**Keywords:** Eucalyptus; Multi-Satellite data; Google Earth Engine (GEE)

## 1. Introduction

Characterizing changes in forested areas is of particular importance to the study of terrestrial and atmospheric carbon circle, as they play a key role in modulating carbon flux between the biosphere and the atmosphere [1–4]. Considerable efforts have been made to monitor the deforestation worldwide in the last several decades, but little has been done to investigate the fast expansion of forest plantations, even though they exert a significant influence compared to the deforestation [5,6]. As reported by the Food and Agriculture Organization of the United Nations (FAO), the area of planted forest has increased notably in all climatic domains over the last 25 years, i.e., it increased respectively by 67% and 51% in the tropical and temperate zones [7]. Tree plantations, especially those established on what was formerly natural forest land, may cause various environmental and social impacts if not managed properly [8–10].

Eucalyptus is one of the most planted genus of tree around the world, particularly in high-yield, intensively managed, short rotation plantations throughout the tropical and temperate zones thanks

to its fast growth and capability to adapt to various habitats [11,12]. It provides not only wood for a wide range of purposes such as timber, pulp, charcoal, firewood and building materials, but also eucalyptus oil that can be used for cleaning, as an antiseptic, and in food supplements [12,13]. Nevertheless, wide concerns have been raised about consumption of soil nutrients, depletion of soil water, biodiversity reduction, and production of allelopathic chemicals of eucalyptus planting [8,14,15]. Some researchers argued that the negative environmental impacts of eucalyptus are mainly because of poor management (i.e., rotation period) rather than its biological characteristics [13]. Despite the socio-economic importance and possible environmental impacts, it is not known, however, what exactly the global or regional area of eucalyptus is [6,12,16].

Recently, several methods have been proposed to map fast-rotation eucalyptus plantations using remote sensing data. For instance, le Maire et al. [6] classified eucalyptus plantations across Brazil by examining whether the Normalized Difference Vegetation Index (NDVI) time series of a pixel is matched with the eucalyptus NDVI reference data. They adopted MODerate resolution Imaging Spectroradiometer (MODIS) 16-day 250 m NDVI time series in their application because the high temporal resolution of MODIS data allows for an accurate computation of the matching. But the moderate spatial resolution is not quite proper to map eucalyptus woodlots owned by small holders that are very common in China, Ethiopia and other countries [12,13]. Mapping eucalyptus at high resolution (30 m) was tested only at small scales [16]. The authors introduced an inverted triangle area methodology to capture the sudden drops in a NDVI time series over 15 years, and classified a pixel as eucalyptus if two such drops were observed by assuming that eucalyptus plantations should experience two harvests during this period. However, the construction of a long and continuous NDVI time series is challenging in many tropical and subtropical regions due to frequent clouds, and the requirement of two entire rotations will fail to detect eucalyptus plantations converted from natural forests or agricultural land in recent years. There is a critical need to to map fast-rotation eucalyptus plantations at high spatial resolution and at large scales.

With the opening of global archives of Earth Observation (EO) data that streamed from modern satellites such as Landsat 8 and the Sentinel series, we have arrived at a new era of global environmental monitoring using remote sensing. The delivery of free satellite imageries captured on a regular basis created huge opportunities for researchers to study the dynamic changes in natural resources [2,5,17]. The task to effectively manage and analyze massive volumes of multi-temporal, multi-satellite datasets over very large areas, however, increasingly challenges the capabilities of personal processing environments [18]. Fortunately, these barriers could be removed by adopting cloud computing platforms such as Google Earth Engine (GEE) and Thematic Exploitation Platforms (TEP) launched by European Space Agency (ESA) [19,20]. Free access to the open geospatial data combined with cloud computing allows researchers to observe the temporal and spatial dynamics of Earth's surface at continental or global scale. Very recently, several studies have used the GEE platform for large scale mapping or monitoring [21–24].

In this paper, we present a new method to produce eucalyptus distribution maps at 30 m resolution of a large province, Guangxi, China. Remotely sensed data from Landsat 8 Operational Land Imager (OLI), MODIS, and Sentinel-2 MultiSpectral Instrument (MSI) are employed to identify short-rotation eucalyptus plantations automatically on the GEE Platform. By studying both the pattern of vegetation indices due to tree crop rotation and spectral characteristics of eucalyptus in the red-edge region, we could produce high resolution maps using a short observation history, i.e., five or six years. The method could capture forest disturbances like conversion of natural forest or agricultural lands to eucalyptus plantations emerged in recent years. In addition, the proposed method allows gaps in the time series, long and continuous observations over the study area is not required as a result.

## 2. Study Area

The study area, Guangxi province, is located in the south of China, extending from 20°54′N to 26°24′N and from 104°28′E to 112°04′E. It consists of 14 administrative regions and covers an area

of 237,600 km$^2$ in total (Figure 1). Thanks to the humid subtropical climate, which is featured with abundant rainfall and warm temperature, this area is rich in plant resources for both food and fibre. As reported by Forestry Bureau [25], the percentage of forest cover in Guangxi reached 62.20% in 2015, ranking the third highest in China. It is the largest wood producing area in China with an annual production of approximately 25 million m$^3$. Eucalyptus (*Eucalyptus camaldulensis*, *Eucalyptus citriodora*), pine (*Pinus massoniana*, *Pinus kwangtungensis*), and fir (*Cunninghamia konishii*) are the main tree species cultivated for timbers, among which eucalyptus accounts for about 70.40% of the total production [25]. Tree plantations belong to either state-owned forest companies or small holders like local peasants. The sizes of farms, as a consequence, vary significantly, ranging from several ares to hundreds of hectares. Other vegetation covers in the study area include evergreen natural forest, croplands for rice and sugarcane, and fruit orchards such as lychee, longan, mandarin, and mango. It is a challenging task to classify land covers or identify a specific species in this region, since the vegetation types are quite heterogeneous and they often exhibit both spectral and seasonal similarities. In addition, because of the homogeneous climatic environment in this area, geophysical data (such as climate, precipitation, and elevation) cannot effectively resolve the confusion among land-cover types [26–28].

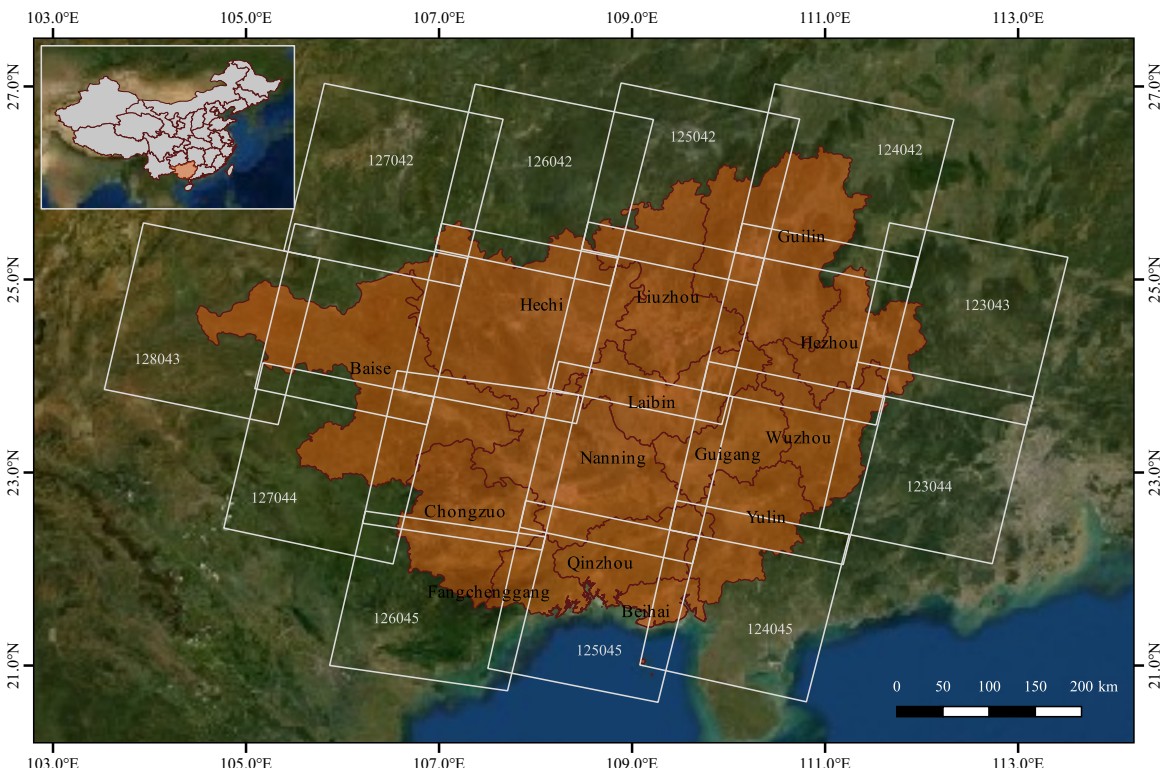

**Figure 1.** Guangxi, China. The study area is located in the south of China. It includes 14 administrative regions and Nanning is its capital. The whole area extends 18 Landsat scene footprints as shown in the figure, where peripheral footprints are labeled using World Reference System (WRS) Path/Row.

## 3. Method

The workflow for identifying eucalyptus plantation is shown in Figure 2 (The whole detection procedure is implemented on GEE, source code can be found at https://code.earthengine.google.com/?accept_repo=users/siatsiatns/eucalypt). Landsat 8 OLI data from year 2013 to 2018, MODIS product MCD43A4 of the same time, and Sentinel-2 MSI imageries from year 2016 to 2018 are adopted, where the MODIS data is employed to validate the quality of Landsat images, i.e., the cloud removal performance, owing to its high repetition rate. The detection procedure is composed of three main steps. First, a time series of Enhanced Vegetation Index (EVI) [29] is constructed from Landsat data for

each pixel. And a statistical hypothesis testing is followed to determine whether the pixel belongs to a tree plantation or not based on the idea that tree crops should be harvested in a specific period. Then, a broadleaf/needleleaf classification is applied to distinguish eucalyptus from coniferous trees such as pine and fir using the red-edge bands of Sentinel-2 data, since eucalyptus is the only broad-leaved species cultivated intensively for timbers in the study area. At last, superpixel based refinements is performed to remove the salt-and-pepper effects resulted from per-pixel detection by taking into account that eucalyptus plantations are usually managed parcel by parcel.

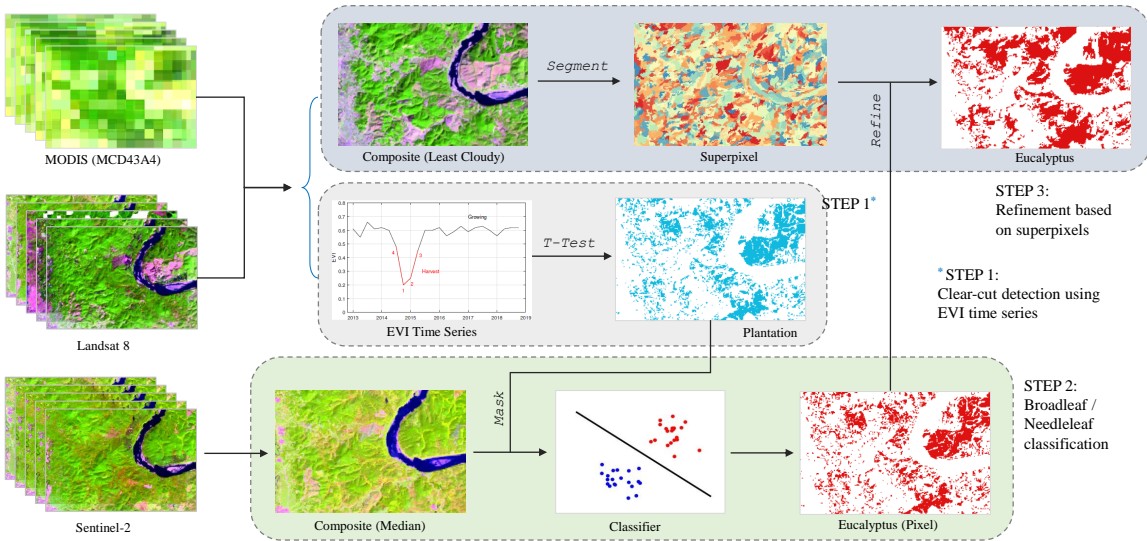

**Figure 2.** The workflow to detect eucalyptus plantations. The whole precedure is accomplished on Google Earth Engine (GEE).

## 3.1. Pre-Processing

Nearly 2000 Landsat 8 OLI images in 18 scene footprints (Figure 1) were processed. The data have been atmospherically corrected using Landsat 8 Surface Reflectance Code (LaSRC) and include a cloud, shadow, water and snow mask produced using CFmask [30,31]. CFmask worked well on most images, but still in some images many cloudy pixels were missed out. We designed a method to eliminate Landsat data with poor cloud removal automatically instead of manually checking a large number of images. The basic idea is to inspect the discrepancy of NDVI derived from Landsat data and simultaneously acquired MODIS images. NDVI values at same locations should be very close if the pixels are not cloudy ones. For each Landsat-MODIS pair, therefore, if a large portion of pixels labeled as *clear* in the Landsat image have very different NDVI values from the MODIS pixels at similar locations, the Landsat image is supposed to have a poor cloud removal result. MODIS Nadir Bidirectional Reflectance Distribution Function-Adjusted Reflectance (NBAR) dataset was employed in the comparison to cancel potential disagreement of surface reflectance retrievals due to orbit difference [32,33]. In addition, NBAR data obtained in 16 consecutive days around the acquiring date of Landsat data were mosaicked to reduce the omission rate of cloud detection in MODIS product and to create a MODIS image as less contaminated by clouds as possible. Figure 3 shows two examples of detected Landsat images in which the clouds are not removed correctly.

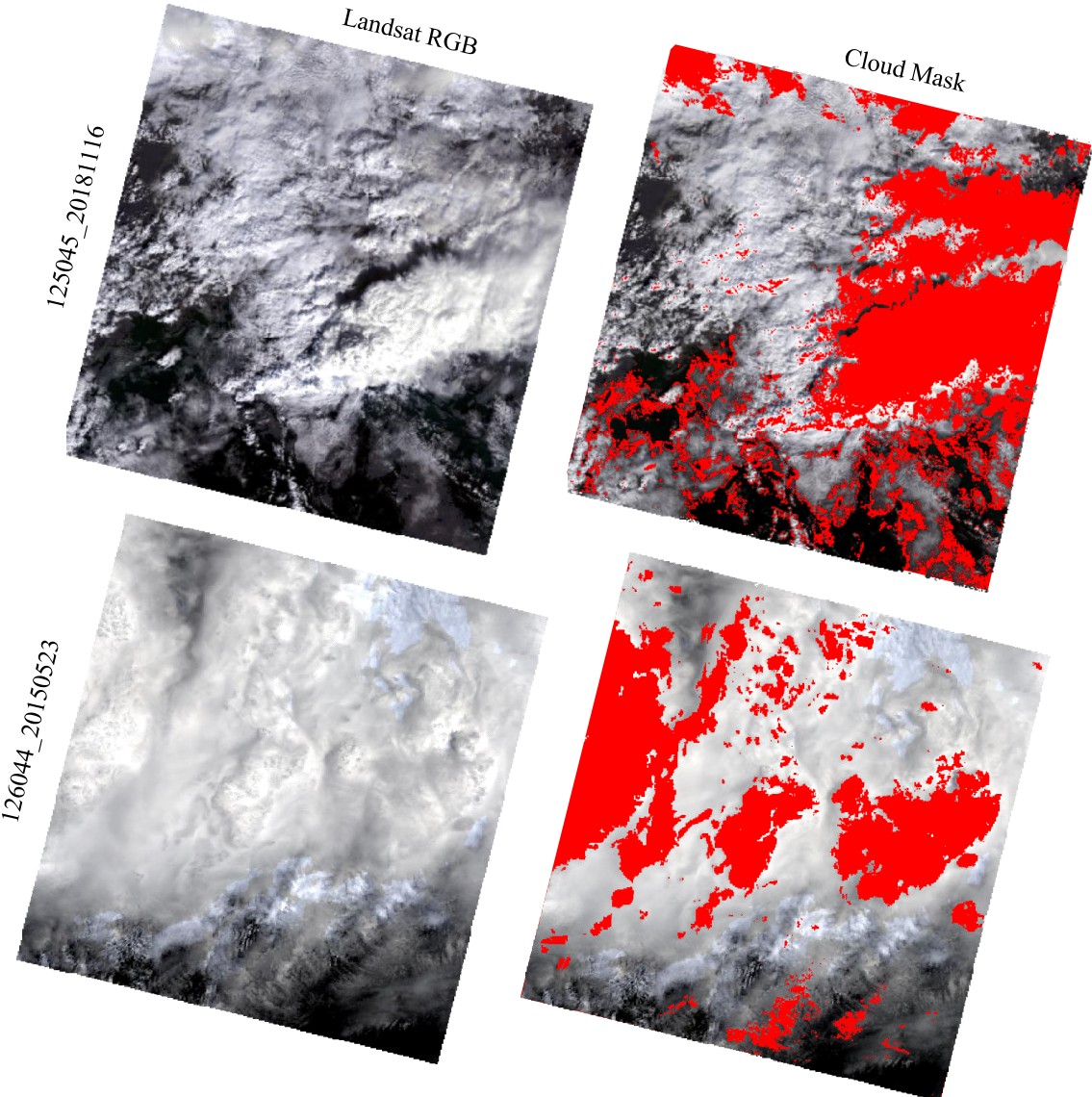

**Figure 3.** Two examples of Landsat data with poor cloud removal results. The first column shows the true color using RGB bands and the second column shows the cloud pixels (labeled as red) detected by the CFmask algorithm.

## 3.2. Clear-Cut Detection

Tree crops for timbers usually have a clear rotation circle, from planting, growing to logging, which lasts from several years to decades. Consider the fast-growing eucalyptus for example. Young trees of 30 cm approximately in height are transplanted to the field after site preparation. They grow slowly in the first months and herbicides are applied at regular intervals to suppress non-crop vegetation such as grasses, vines, and shrubs. Once new roots are developed, the trees begin to flourish and have a rapid increase in foliage area. They keep growing in the next four or five years, at a rate of 1.5–3 m in height per year. The plantation maintains a dense vegetation cover all through these years until the trees are harvested, when it becomes bare soil for several weeks due to logging and herbicide application for the next rotation circle. Rotation length for the fast-growing eucalyptus is usually 5–6 years, but it may range from 3 to 8 years. We utilized this phenology information to identify tree plantations from natural forests, orchards, and croplands.

Vegetation indices, EVI specifically, were employed to capture the change of vegetation cover in a period of 5–6 years. Compared with NDVI, EVI is more sensitive in high biomass regions and is less influenced by atmosphere conditions [29]. It is computed as follows:

$$EVI = 2.5 \times \frac{\rho_{nir} - \rho_r}{\rho_{nir} + 6.0\rho_r - 7.5\rho_b + 1} \tag{1}$$

where $\rho_{nir}$, $\rho_r$, and $\rho_b$ stand for the atmospherically-corrected surface reflectance of the near-infrared, red (visible) and blue (visible) bands of Landsat data. Figure 4 shows several examples of EVI time series obtained from Landsat data at an interval of 16 days (the repetition interval of Landsat 8). Due to frequent cloud contamination, plenty of pixels were masked out and the EVI values were inaccessible, leaving lots of gaps in the series. To avoid these gaps and reduce residual noise, the median of EVI values for each period of 3 months were computed to form another time series, shown as red circles in Figure 4. Tree plantation detection was based on the new time series.

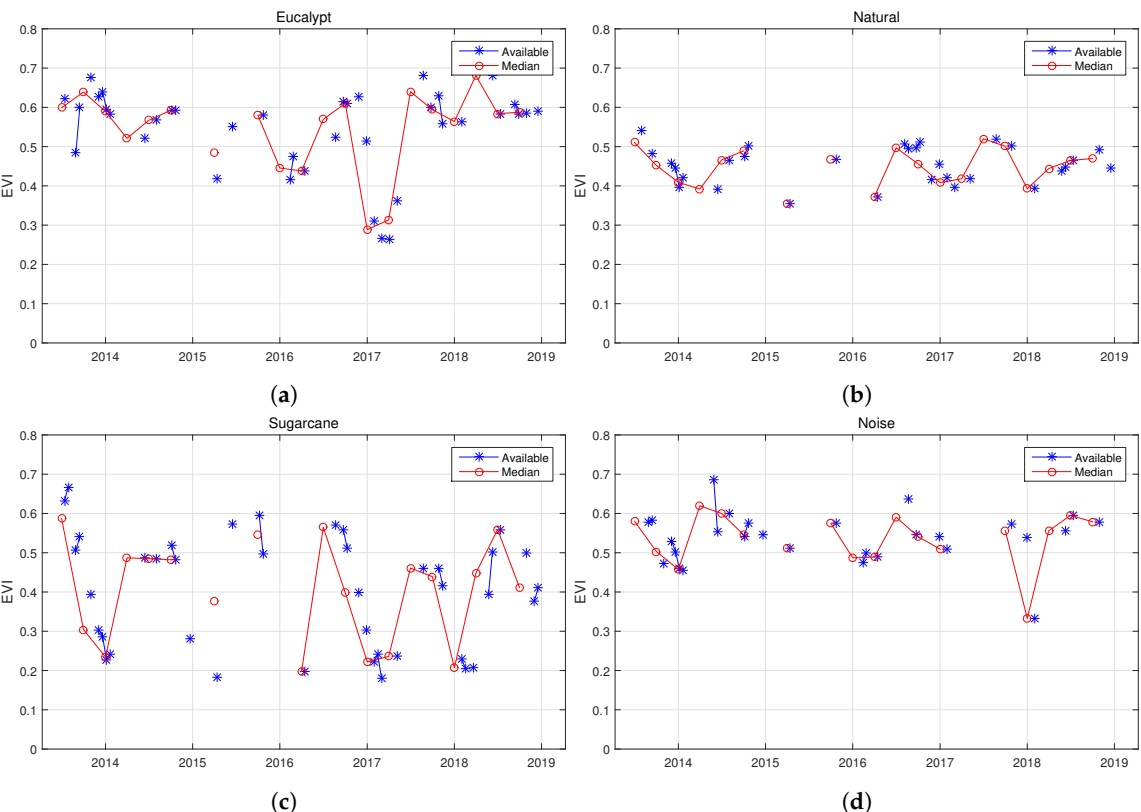

**Figure 4.** EVI time series examples including (**a**) tree plantation (eucalyptus), (**b**) cropland (sugarcane), and (**c**) natural forest from 2013 to 2018. The time series marked with blue asterisks is obtained from Landsat data at an interval of 16 days, whereas the other one marked with red circles is constructed from median EVIs in each period of 3 months. The last example (**d**) shows a noise case where the EVI series has a sudden drop and rise.

Clear-cutting of trees over a plantation gives rise to abrupt drops in the EVI time series. A detector based on statistical analysis to catch the sudden change of EVI in 5–6 years was designed. To start with, each series was segmented into two parts, with one representing the *growing* stage, and the other the *harvest* stage. The *growing* part is supposed to have much more and larger EVI values than the *harvest* part. In addition, the *harvest* part must be consecutive in time with a maximum duration of 1 year. The segmentation was accomplished iteratively, starting from extracting the least value as the *harvest*

part, and then extending it by ingesting the adjacent values, as illustrated in Figure 5. A neighboring value was added to the *harvest* part if it decreased the Sum of Squared Error (SSE) defined as:

$$SSE = \sum_{i=1}^{N_h} (h_i - \mu_h)^2 + \sum_{i=1}^{N_g} (g_i - \mu_g)^2 \qquad (2)$$

where $h_i$ and $g_i$ represent the values in the *growing* part and the *harvest* part respectively, $N_h$ and $\mu_h$ ($N_g$ and $\mu_g$) are the total number of values as well as the mean value of the *harvest* part (the *growing* part).

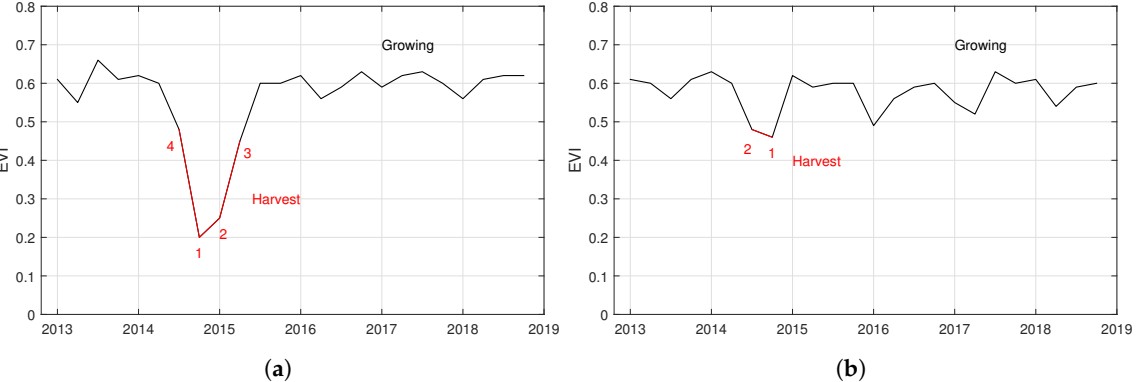

(**a**)　　　　　　　　　　　　　　　　　　　　(**b**)

**Figure 5.** Illustrations of time series segmentation. Each time series (with or without clearcuts) is divided into a *Harvest* part (red) and a *Growing* part (black). As labeled by the numbers, the *Harvest* part is sliced off from the time series iteratively starting from the least value with the ojbective to reduce the Sum of Square Error (SSE). (**a**) An example of tree clearcut. (**b**) An example of no clearcut.

A statistical hypothesis testing was then followed to determine whether the two parts had significant differences in mean values [34]. Suppose that the clear-cut would cause the average of EVI to decrease by at least *d*, a one-tailed t-test was conducted, with the null hypothesis $H_0$ (the two parts have significant difference) and the alternative hypothesis $H_1$ (the difference is not large enough) stated as

$$H_0 : \mu_g - \mu_h \leq d$$
$$H_1 : \mu_g - \mu_h > d$$

as well as the test statistic given by

$$t = \frac{\mu_g - \mu_h - d}{\sqrt{\frac{\sigma_h^2}{N_h - 1} + \frac{\sigma_g^2}{N_g - 1}}} \qquad (3)$$

where $\sigma_g$ and $\sigma_h$ are the standard deviation of the *growing* part and the *harvest* part. In the approach, *d* was set to 0.12. The choice of a proper *d* is discussed in more details in Section 5. The null hypothesis was rejected if *t* was larger than the critical value corresponding to a specific significance level, meaning that $\mu_g$ was significantly larger than $\mu_h$. We could state that clear-cut logging has taken place, and the pixel under analysis was assumed to belong to a tree plantation.

To summarize, the objective of the above procedure is to find out pixels where the EVI time series can be divided into two parts with very different mean values, by means of time series segmentation and statistical hypothesis testing. In order to make sure those pixels belong to tree plantations, constraints such as maximum length of the *harvest* part (1 year) and minimum length of the whole time series are added. To reduce the residue noise, a time series consisted of median values in each 3 month period is created. In addition, the hypothesis testing is performed only when the number of values in the *harvest* part is larger than 1 to eliminate the sudden drop caused by noise as shown in

Figure 4d. The EVI time series is not required to be continuous, which allows the detection procedure to work in areas covered by clouds frequently, the tropical and subtropical regions for instance.

### 3.3. Broadleaf/Needleleaf Classification

As mentioned above, eucalyptus, pine, and fir are main species cultivated in tree plantations in the study area. Although the rotation length is different, 5–6 years for eucalyptus, 10–12 years for fir, and 18–20 years for pine, the previous procedure could detect out not only eucalyptus plantations but also part of fir and pine plantations using observation data of 5 or 6 years. We need further to distinguish the eucalyptus from both pine and fir. Considering that eucalyptus is the only broadleaf tree among the three species, a binary classification between broad-leaved trees and coniferous ones was implemented using Sentinel-2 data. Differences of the reflectance values in the red-edge spectral region have been observed among various vegetation covers [35–38]. It was reported that the reflectance of the red-edge and SWIR bands were considered the most important ranked attributes for the classification of Sentinel-2 data [39]. We also found that the red-edge bands of Sentinel-2 data for the three tree species were different, i.e., the eucalyptus had larger values in these bands than the pine and the fir do, as shown in Figure 6.

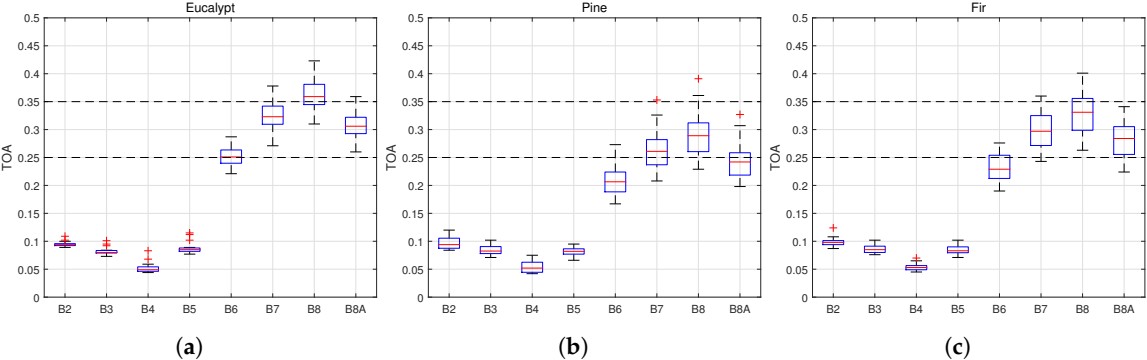

**Figure 6.** Boxplots of the Top of Atmosphere (TOA) reflectance of (**a**) eucalyptus, (**b**) pine, and (**c**) fir samples from Sentinel-2 data. Image bands covering the visible (B2, B3, B4), the red-edge (B5, B6, B7, B8A), and the near-infrared (B8) region are compared.

Support Vector Machine (SVM) was adopted as the classifier, which is particularly appealing in the remote sensing field due to its ability to generalize well even with limited training samples [40,41]. In order to remove cloudy pixels as well as to avoid residue noise due to weather conditions, median composites were generated from Sentinel-2 images acquired in the peak growing season (from May to September). Top of Atmosphere (TOA) reflectance of the red-edge bands (B5, B6, B7, B8A) and the near-infrared band (B8) were fed to the classifier directly without atmospheric correction. The classifier was first trained on 1965 samples which had been identified visually through high resolution images from Google Earth. The overall accuracy of a test on another 2320 samples achieved 90.77%, demonstrating that the combination of SVM with the selected bands is capable to separate broadleaf and needleleaf trees. The trained classifier was then used to extract eucalyptus plantations from pixels labeled as tree plantation after previous step. The Sentinel-2 data was re-sampled into 30 m resolution automatically on GEE to match the detection result from Landsat data. We processed Sentinel-2 data from year 2016 to year 2018 to ensure that the plantations were covered by developed trees.

### 3.4. Refinements Based on Superpixels

Pixel based detection may lead to salt-and-pepper effects in the maps, i.e., holes in large blocks of identified pixels as well as isolated small eucalyptus pixel groups. This is because the pixels are processed independently, regardless of the contextual information. Provided that the knowledge of a pixel belonging to a certain class increases the probability that its neighboring pixels belong

to the same class, it is widely accepted that integrating spatio-contextual information with spectral information could improve the accuracy of applications such as land cover classification and thematic mapping [17,42–44]. Since eucalyptus plantations are usually managed parcel by parcel, and each parcel appears as a homogeneous region, it is necessary to take into consideration of this contextual knowledge in the detection. Image segmentation is a common way to extract contextual information. In our approach, a superpixel segmentation algorithm named Simple Non-Iterative Clustering (SNIC) was employed, which is fast, requires little memory, and achieves the state-of-the-art performance [45].

Segmentation can be incorporated into the detection procedure either when generating EVI time series or as a post-processing step. In the former way, images are segmented and spectral characteristics of superpixels are used for clear-cut detection instead of those of pixels, as the standard manner of Object-Based Image Analysis (OBIA) [42]. However, merging segments obtained from images in the time series could result in myriads of small fragments since each image will produce a different segmentation. In this approach, we adopted the latter way. SNIC segmentation was applied to a composite of Landsat images. A winter composite was created in order to obtain clear boundaries between croplands and forest areas, where images with fewer cloudy pixels had higher priority during the mosaicking. For each segment or superpixel, all pixels were assumed to belong to eucalyptus plantations if more than 30% of pixels were identified as eucalyptus ones (refer to Section 5 for more details about the threshold). In addition, isolated small groups of detected pixels were discarded. The choice of segmentation scale impacts the final performance. However, there is no standardized or widely accepted method to determine the optimal scale for various applications, areas with different environmental and biophysical conditions, and different kinds of remotely sensed images [43]. We initialized the segmentation with a $10 \times 10$ grid (in pixel), and further subdivided a segment using $5 \times 5$ grids if the variance of its pixels is larger than the median of all variances (calculated from all superpixels). This could cut small plantations as well as large ones appropriately.

## 4. Result and Assessment

Figure 7a shows the extent of short-rotation eucalyptus plantations detected by the proposed method. Eucalyptus plantations are mainly distributed in the central and the south of the study area, in administrative districts such as Laibin, Nanning, and Qinzhou. There are large blocks of detected pixels representing plantations managed by state-owned forest enterprises (e.g., in Nanning and Laibin) as well as small clusters representing patches hold by local peasants and small companies (e.g., in Qinzhou). Eucalyptus area is calculated for each administrative district, see Table 1. The amount is quite different, ranging from 35,009 ha (Guilin) to 210,990 ha (Nanning). The total area of eucalyptus plantations reaches 1,439,221 ha, nearly 5.54% of the land area.

We evaluated the detection result using two different data sources. One is land cover samples acquired during a field survey in 2017, and the other is eucalyptus plantations visually identified based on very high resolution images from Google Earth. The field trip lasted for almost a month, and a total number of 20,327 samples were collected. Each sample was labeled with one of the six land cover categories including forest, grassland, cropland, man-made surface, wetland, and water bodies. Meta information such as collecting date, photo, and simple description was also recorded. For forest and cropland samples, we added a feature describing the vegetation cover, i.e., eucalyptus, pine, natural forest, sugarcane, and so on. Only samples with identified vegetation cover were adopted in our assessment, and there were 12,117 of them in total, see Figure 8a. The error matrix and evaluation metrics based on these samples are shown in Table 2 (the first tabular). The overall accuracy achieves 87.97%, and the producer's accuracy as well as the user's accuracy of eucalyptus is, respectively, 63.85% and 66.89%. For evaluation based on high resolution data, we selected 19 regions of size $4.45 \text{ km} \times 4.45 \text{ km}$ on a regular grid, and obtained the reference data via visual interpretation of Google Earth images, see Figure 8b. All the 19 regions, 418,546 pixels in total, were evaluated together, and the error matrix as well as accuracy metrics are shown in Table 2 (the second tabular). The overall accuracy reaches 93.59%, which is about 5.62% higher than that in the previous assessment owning

to the increase of the detection accuracy for non-eucalyptus pixels. The producer's accuracy in both assessments demonstrates that the proposed method tends to underestimate eucalyptus plantations. There are many reasons for that, but the lack of data due to frequent cloud and the environmental policy to restrict forest logging in recent years are the two most crucial ones. This will be explained in more details in Section 5.

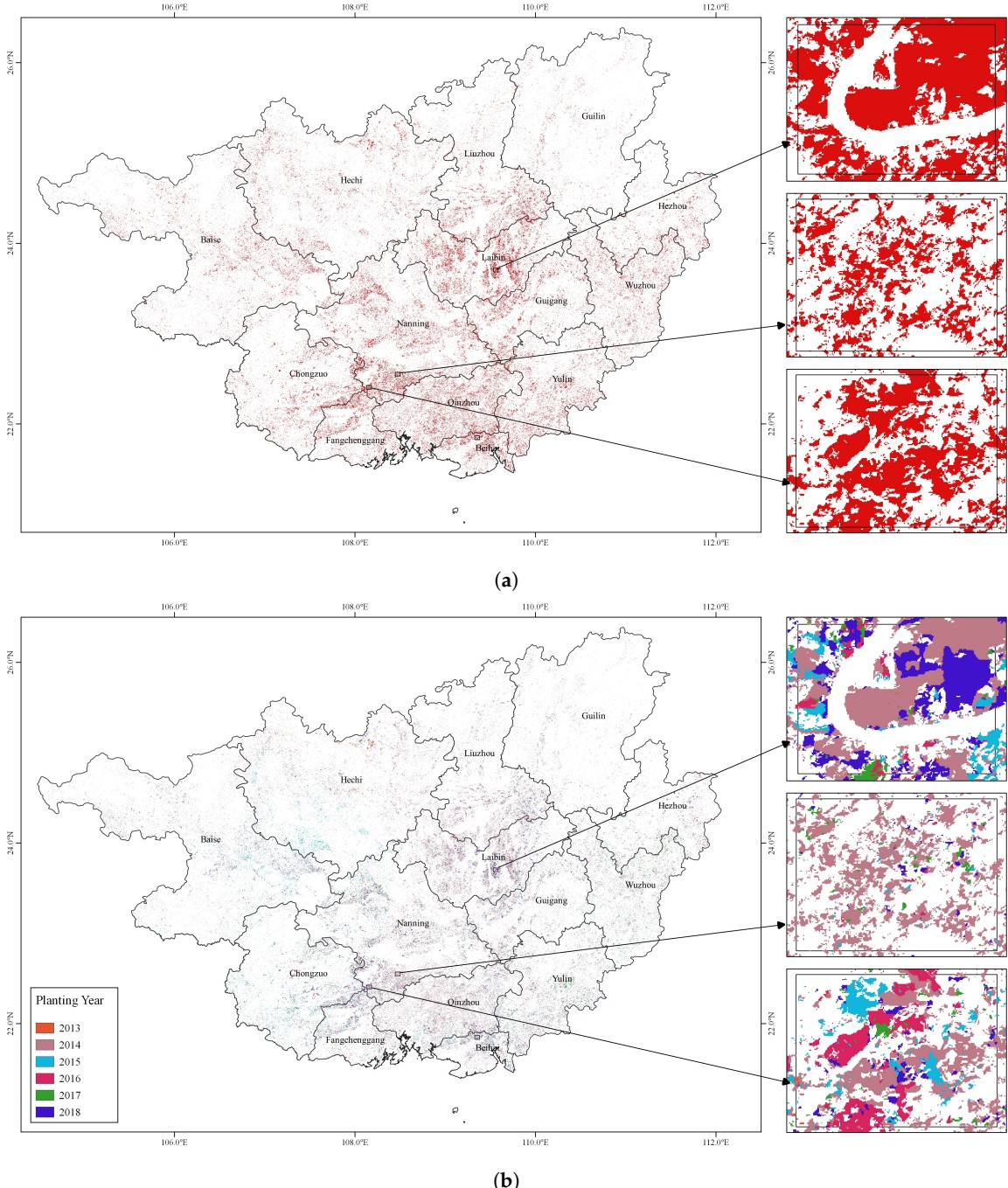

**Figure 7.** Distribution of eucalyptus plantations in Guangxi. (**a**) Detected extent of short-rotation eucalyptus plantations. (**b**) The planting calendar of eucalyptus.

**Table 1.** Area of eucalyptus plantations in Guangxi.

| District | Area (ha) | % of Land Area |
|---|---|---|
| Baise | 123,659.73 | 3.10% |
| Beihai | 44,619.39 | 11.88% |
| Chongzuo | 98,650.98 | 5.22% |
| Fangchenggang | 57,632.76 | 8.86% |
| Guigang | 78,898.50 | 6.79% |
| Guilin | 35,009.64 | 1.14% |
| Hechi | 124,611.66 | 3.38% |
| Hezhou | 45,170.10 | 3.51% |
| Laibin | 163,282.95 | 11.30% |
| Liuzhou | 100,117.26 | 4.80% |
| Nanning | 210,990.60 | 8.74% |
| Qinzhou | 150,095.16 | 12.91% |
| Wuzhou | 87,910.92 | 6.45% |
| Yulin | 118,571.85 | 8.48% |
| Total | 1,439,221.50 | 5.54% |

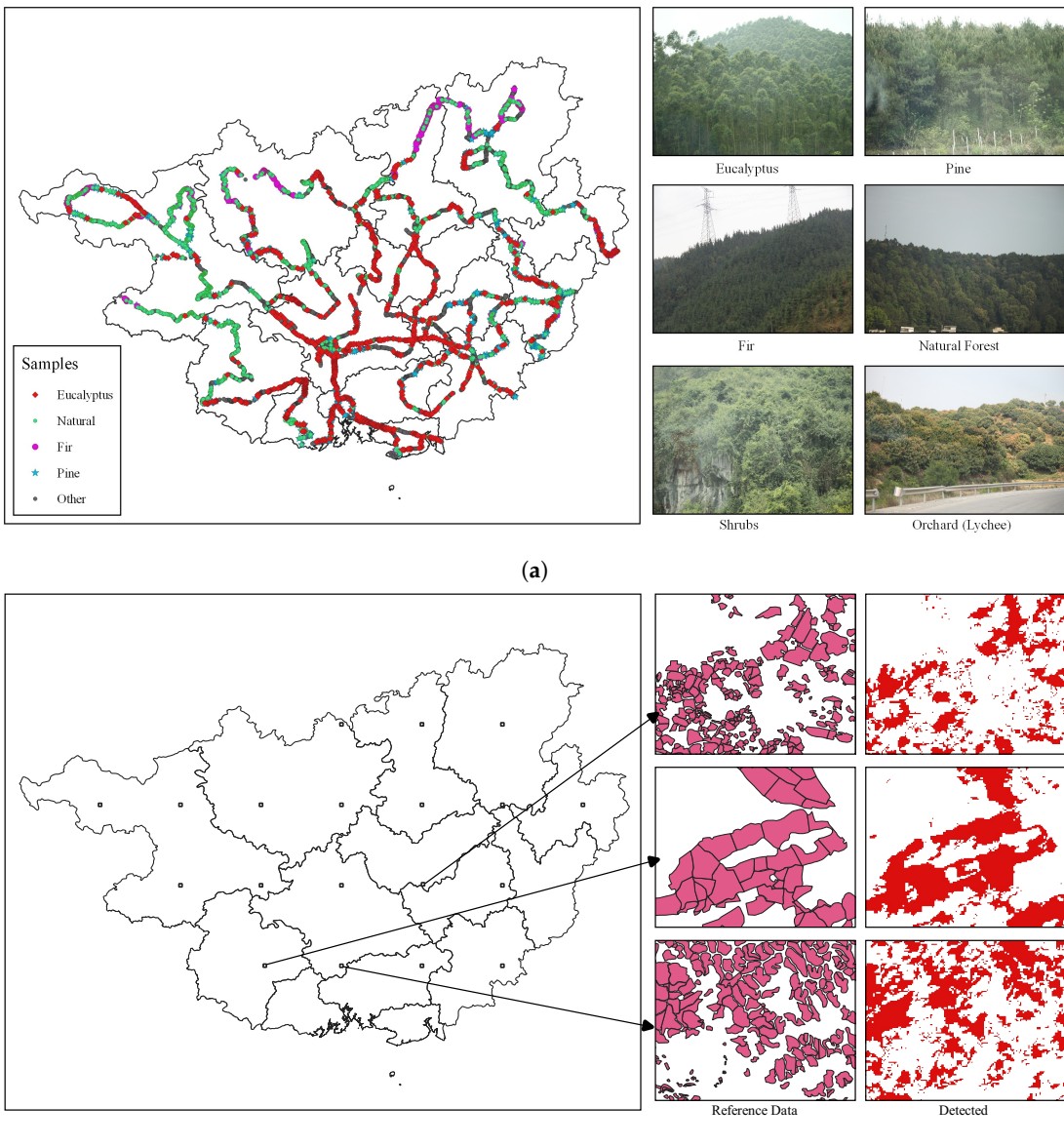

**Figure 8.** Two different data sources used to evaluate the detection result. (**a**) Samples collected in a field survey. (**b**) Reference data identified using Google Earth images in 19 regions.

As a comparison, we conducted an experiment to detect eucalyptus plantations by applying a multi-class SVM to Landsat data and Sentinel-2 data directly. A summer image and a winter image were composited and fed to the classifier instead of adopting all the Landsat images, along with the Sentinel-2 median composite (see Section 3.3). Pixels which have small NDVI values in both the summer image and the winter image were considered as water bodies or built-up areas and were masked out before classifying. Besides the samples of eucalyptus, pine and fir we used to train the binary classifier in Section 3.3, samples of natural forest, shrubs, lychee, sugarcane, and rice were also added to train the multi-class SVM classifier. Accuracy assessment of the SVM output using field survey samples is shown in Table 2 (the last tabular). We can see that the proposed method outperforms the SVM on all metrics including the overall accuracy, the producer's accuracy and the user's accuracy. The evaluation of SVM classifier using high resolution data is even worse, and the result is not listed here. We noticed that the multi-class SVM classifier is prone to confuse eucalyptus with natural forest and shrubs, and gives a very low user's accuracy for eucalyptus plantations as a consequence.

**Table 2.** Accuracy assessment. Confusion matrix as well as accuracy metrics such as producer's accuracy, user's accuracy and overall accuracy (emphasized in bold face) are calculated. Evaluation of proposed method using two different data sources (samples obtained in a field trip, and ROIs interpreted manually interpreted), as well as evaluation of SVM using field samples are compared.

| Samples | Eucalypt (Detected) | Non-Eucalypt (Detected) | Producer's Accuracy |
|---|---|---|---|
| Eucalypt (Reference) | 1374 | 778 | 63.85% |
| Non-Eucalypt (Reference) | 680 | 9285 | 93.18% |
| User's Accuracy | 66.89% | 92.27% | **87.97%** |
| **ROI** | Eucalypt (Detected) | Non-Eucalypt (Detected) | Producer's Accuracy |
| Eucalypt (Reference) | 22547 | 14946 | 60.14% |
| Non-Eucalypt (Reference) | 11876 | 369177 | 96.88% |
| User's Accuracy | 65.50% | 96.11% | **93.59%** |
| **SVM** | Eucalypt (Detected) | Non-Eucalypt (Detected) | Producer's Accuracy |
| Eucalypt (Reference) | 971 | 1181 | 45.12% |
| Non-Eucalypt (Reference) | 2489 | 7476 | 75.02% |
| User's Accuracy | 28.06% | 86.36% | **69.71%** |

A eucalyptus planting calendar was also produced, where the planting date was obtained from the date when the EVI reaches the minimum value. Due to frequent cloud interference and relatively low repetition rate of Landsat 8, it is impractical to retrieve the planting date to day or month. We created the planting calendar on a yearly basis, and ignored the offset from the actual planting date to the date corresponding to the least EVI. The result is shown in Figure 7b and the planting area of each year is calculated in Table 3. It demonstrates that the planting area of eucalyptus reaches a peak in 2014, accounting for almost 39.26% of the total area planted through 2013 to 2018. A further review of Landsat data and high resolution Google Earth images revealed that a large number of sugarcane fields were converted to eucalyptus plantations in 2014, see Figure 9 for an example in Qinzhou. In addition, the Landsat 8 data archive in 2014 over the study area is continuous, but not the same case for other years. Take year 2013 for example, not until July did Landsat 8 begin to acquire data in some regions, making the proposed method fail to capture the clear-cut happened in this year and leading to underestimation of eucalyptus area.

**Table 3.** Area of eucalyptus by year.

| Year | Area (ha) |
|------|-----------|
| 2013 | 155,538.91 |
| 2014 | 565,096.22 |
| 2015 | 239,765.04 |
| 2016 | 129,719.43 |
| 2017 | 154,069.11 |
| 2018 | 195,032.79 |

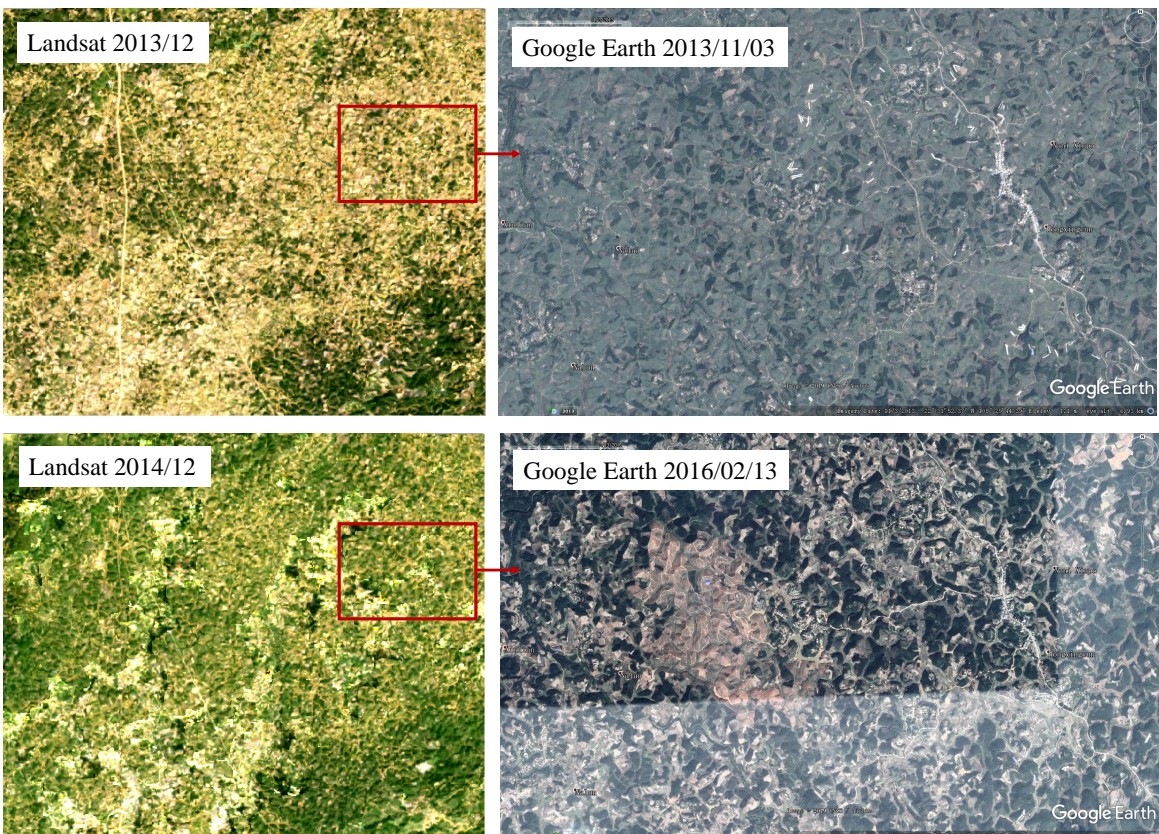

**Figure 9.** Landsat data and Google Earth images before and after year 2014. Combination of Landsat visible bands shows that there is a significant increase of green areas in 2014. High resolution Google Earth images also confirms that large number of cropland fields are converted into eucalyptus plantations. Here Google Earth data at early 2016 is compared because there is no data covering this area in 2014 and 2015.

## 5. Discussions

As mentioned before, there are two main reasons some eucalyptus plantations are not detected out by the proposed approach. First of all, the lack of observation data due to frequent cloud causes our method to miss the clear-cut of eucalyptus which usually takes place in a short window of time. Even if we adopt all the images in the Landsat 8 data archive, generating a continuous time series at an interval of 3 months is still impossible for many pixels. The problem of cloud is inevitable in tropical and subtropical zones when using optical remote sensing data. Fortunately, this situation will be improved in the future by fusing data from different sensors as the launch of Sentinel-2 satellites which have a similar spatial resolution as Landsat 8 in many bands and a higher repeating rate (6 days with both Sentinel-2A and Sentinel-2B). Synthetic Aperture Radar (SAR) could be another option to tackle the cloud problem thanks to the cloud penetration capability of microwaves. Secondly, the curtailing quota of lumber logging due to more and more stringent environmental policies in recent years also

has a great impact on the result. For instance, large eucalyptus plantations near water sources are not allowed to cut down in many districts, and the changing pattern of EVI time series over eucalyptus plantations disappears. In addition, the delay of logging license postpones the harvest of eucalyptus which may not be captured using Landsat 8 observation for 5 or 6 years. A possible solution is to incorporate high resolution data and explore the texture information of eucalyptus plantations.

Usually, there is a bare soil zone of approximate 1 m in width to separate eucalyptus into row by row, which will reduce the actual vegetation indices of eucalyptus to some extent. In this paper, we applied techniques such as time series segmentation and statistical hypothesis test to capture the abrupt changes in time series, instead of inverting bio-parameters quantitatively based on EVI. So this influence could be neglected.

A detection task usually involves the strategies to balance the error rate of omission and the error rate of commission. In our approach, the value of $d$ in (3) needs to be considered carefully, a large $d$ will fail to identify many eucalyptus plantations (high omission error rate) while a small value will misclassify non-eucalyptus pixels to eucalyptus ones easily (high commission error rate). We compared the two types of error rate by selecting different values for $d$. As shown in Figure 10a, the omission factor rises steadily as we increase $d$ whereas the commission factor follows an opposite trend. The addition of both error rates manifests a U-shaped curve, which reaches the nadir when $d$ is equal to 0.12. The choice of $d$ depends on what level of omission error or commission error is demanded. In this work, we chose $d = 0.12$ to minimize the sum of both error rates. A similar analysis of the threshold in Section 3.4 is also performed, as shown in Figure 10b.

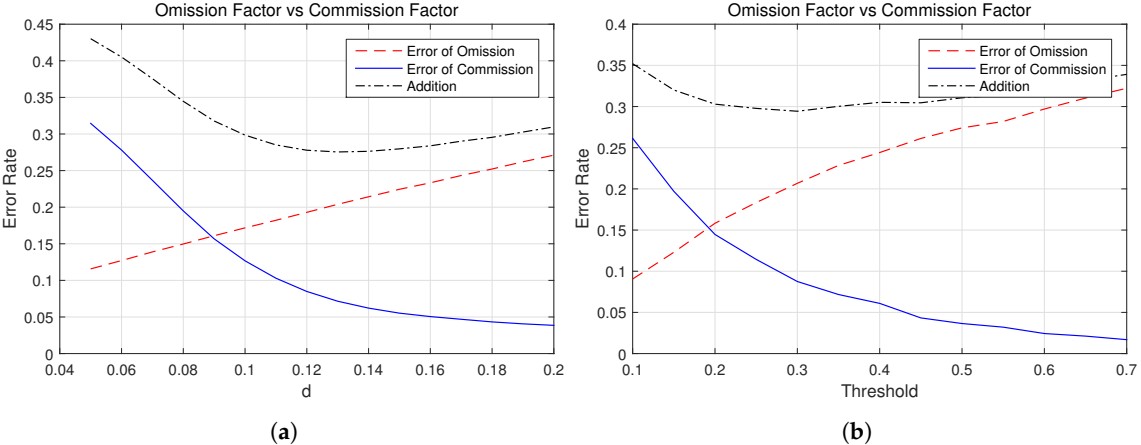

**Figure 10.** Omission factor vs commission factor. (**a**) The change of error rates of eucalyptus detection as varying $d$ in (3). (**b**) The change of error rates of eucalyptus detection as varying the threshold in Section 3.4.

## 6. Conclusions

High-yield, intensively managed, short rotation plantations, also known as fast-wood plantations, are becoming more and more widespread in South America, Asia and other areas in the latest decades. The environmental and social impacts of this type of plantations has caused them to become controversial. For instance, short-rotation eucalyptus plantations are often criticized for the consumption of soil nutrients, depletion of soil water, biodiversity reduction and so on. The detection of the extent of these plantations is a prerequisite for the study of their influence on ecosystem dynamics as well as on land use/land cover.

It is a difficult task to classify or identify tree species using remote sensing data. Possible solutions include exploring the texture information in very high resolution images where different tree spacing and tree crown sizes can be observed, or the spectral dissimilarity of different tree species at certain seasons using hyperspectral data. However, either high resolution data or hyperspectral data is costly and hard to obtain, which restricts their application at large scales. In this paper,

we presented an approach to identify short-rotation eucalyptus plantations by considering both the phenological information of eucalyptus planting and spectral characteristics of eucalyptus trees in specific wavelengths. Owning to the use of open remotely sensed images from several modern satellites and the free GEE cloud computing platform, the method can be used to produce maps of eucalyptus extent at high spatial resolution (30 m) and large scales. The evaluation of detection result in Guangxi province, the largest wood producing area of China, demonstrates that the proposed approach has a good performance. The procedure can be extended to other eucalyptus planting areas such as Vietnam, India, and Indonesia.

Nevertheless, the approach tends to underestimate the eucalyptus area because of the lack of optical remote sensing data in tropical and subtropical zones, and changing cultivation practice due to economic and environmental policies. A possible solution to this problem in the future could be fusing Sentinel-2 data with Landsat data. The thresholds used in our method are not specified empirically, but determined by balancing the rate of commission error and the rate of omission error. Principally, they should be recalculated on different datasets. Two data sources, land cover samples acquired during a field survey and eucalyptus plantations visually identified based on very high resolution images from Google Earth, are employed to evaluate the proposed method. It is worth mentioning that most field samples are collected along the roads (they are easy to access), and the accuracy of manually identified eucalyptus is subject to the interpreter. So the accuracy of the assessments is more or less biased.

Geophysical data such as precipitation, temperature and elevation are reported to be useful to improve the performance of land-cover classification. In this work, we didn't take them into account because transport is often an important factor in selection of plantation locations, and most eucalyptus are planted in relatively low elevations in Guangxi. Geographical information such as DEM may not help significantly in this case. In other regions with large difference in elevation or slope, adding up DEM could be a useful way to improve the accuracy which should be investigated further.

**Author Contributions:** X.D. designed and performed the experiments, analyzed the data, and wrote the paper; S.G., L.S. and J.C. provided valuable suggestions to write the paper, and revised the paper. All authors have read and agreed to the published version of the manuscript.

**Funding:** This research received no external funding.

**Acknowledgments:** This work was partially supported by the Strategic Priority Research Program of the Chinese Academy of Sciences (Grant No. XDA19030301), and the National Natural Science Foundation of China (Grant No. 41801358).

**Conflicts of Interest:** The authors declare no conflict of interest. The founding sponsors had no role in the design of the study; in the collection, analyses, or interpretation of data; in the writing of the manuscript, and in the decision to publish the results.

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
