# Peer review of "Identification of Short-Rotation Eucalyptus Plantation at Large Scale Using Multi-Satellite Imageries and Cloud Computing Platform"

_remotesensing, doi:10.3390/rs12132153_

Round 1

Reviewer 1 Report

The manuscript is well organized to identify Eucalyptus plantation with high spatial resolution at large scale using multi-satellite imageries and cloud computing platform. There are three minor comments below:

Figure 5: What are (a) and (b)?

Figure 8: "The first one" should be "(a)" and "the other" should be "(b)".

Table 3: Could you compare your estimations with actual data?

Reviewer 2 Report

See attached file

Reviewer 3 Report

Reviews (chapters to improve)

Study area

It stated that as reported by Forestry Bureau [25], the percentage of forest cover in Guangxi reached 62.20% in 2015, ranking the third highest in China. No reference is made on the existing of evaluation of risk maps of the land use for studying the expected heterogeneity of the zone and that the Eucalyptus dominates (in %?) the entire area concerned. Additional existing environmental indexes maps (i.e. to identify the distribution of vegetation) as a combination of remote sensing data (e.g. GIS maps) could help the evaluation also of the ecosystem conditions.

Method

The workflow developed for identifying Eucalyptus plantation doesn’t consider parcels which makes the area a homogeneous zone. The creation of a DEM is also recommended to obtain a robustness of the accuracy assessment. Triangular Irregular Network (TIN) defining cell size as 10 meters in accordance with the spatial resolution of the Sentinel-2A image could be an asset as a digital representation of the topographic area under Eucalyptus plants.

Discussion

Often the Eucalyptus area is planted with a distance between them and this create an area of bare soil between them. This aspect on the influence of intercalated bare soil zones may made the NDVI values lower than the applied in this study. This aspect should be also better assessing in the article.

Conclusions

Consequently, the analysis of the combination of geographical maps (including the geographical information of the DEM, such as elevation range and slopes variation) will make possible the comprehension and a better analysis of the area with presence of Eucalyptus homogenously plants. This aspect should be described in the conclusions to allow news studies and new perspectives of the phenomena, also make possible different studies ways and conclusions.

Finally, the study should contain also elements to understand the ecosystem dynamics in particular for a global analysis of the land use and the confirmation that the Eucalyptus dominates all the area interested, perceive the zones where Eucalyptus trees have more photosynthetic activity and the zones with more eucalyptus density. Evaluating this approach will improve the quality in this study.

Round 2

Reviewer 2 Report

Accepted

Author Response

Thank you again for your advices. 

Reviewer 3 Report

"Author's Conclusions" should much better and more focused with the assessment and the discussions of the study. 

Author Response

Thank you very much for your advice. We reorganized the section of Conclusions, and added a paragraph to summarize the assessment as well as its limitations. In addition, a brief conclusion of the key points mentioned in Discussion is also given. See the third paragraph of Section 6 (line 334 to 345).